# Extracellular Vesicles—A New Potential Player in the Immunology of Renal Cell Carcinoma

**DOI:** 10.3390/jpm12050772

**Published:** 2022-05-10

**Authors:** Marcin Kleibert, Miłosz Majka, Klaudia Łakomska, Małgorzata Czystowska-Kuźmicz

**Affiliations:** 1Laboratory of Centre for Preclinical Research, Department of Experimental and Clinical Physiology, Medical University of Warsaw, Banacha 1b St., 02-097 Warsaw, Poland; marcin.kleibert@gmail.com (M.K.); miloszmajka98@gmail.com (M.M.); 2Chair and Department of Biochemistry, Medical University of Warsaw, Banacha 1 St., 02-097 Warszawa, Poland; 3Faculty of Medicine, Wroclaw Medical University, Wybrzeże L. Pasteura 1 St., 50-367 Wrocław, Poland; klaudia.lakomska@onet.eu

**Keywords:** renal cell carcinoma, extracellular vesicles, exosomes, cancer immunology, cancer-induced immunosuppression, diagnosis, prognostic factor

## Abstract

The incidence of renal cell carcinoma (RCC) has doubled in the developed world within the last fifty years, and now it is responsible for 2–3% of diagnosed cancers. The delay in diagnosis and the not fully understood pathogenesis are the main challenges that have to be overcome. It seems that extracellular vesicles (EVs) are one of the key players in tumor development since they ensure a proper microenvironment for the tumor cells. The stimulation of angiogenesis and immunosuppression is mediated by molecules contained in EVs. It was shown that EVs derived from cancer cells can inhibit T cell proliferation, natural killer lymphocyte activation, and dendritic cell maturation by this mechanism. Moreover, EVs may be a biomarker for the response to anti-cancer treatment. In this review, we sum up the knowledge about the role of EVs in RCC pathogenesis and show their future perspectives in this field.

## 1. Introduction

Renal cell carcinoma (RCC) accounts for 2–3% of global cancer diagnoses [1]. RCC is cancer that originates from the renal epithelium and is responsible for >90% of cancers located in the kidney [2]. In 2021, kidney cancer was the sixth most frequently diagnosed cancer in men and the eighth in women in the United States. Comparing the data from 2018 and 2021, we can observe a decreasing trend in mortality, with the rate declining by 21% (22.9% in 2018 vs. 18.1% in 2021) [3,4]. This decreasing trend can be connected with better diagnostics of patients with early stages of RCC. Additionally, the introduction of targeted therapy like anti-VEGF antibodies or immune check-point inhibitors contributes toward a prolonged overall survival of patients with advanced stages of RCC [5,6,7]. Despite this decreasing trend, RCC is still an enormous challenge for medical oncology.

RCC is divided into three major histological subtypes: clear cell RCC (ccRCC), papillary RCC (pRCC), and chromophobe RCC (chRCC). Clear cell RCC is the most frequent type of RCC (responsible for 75% of cases). The second most common type of RCC is p-RCC (15%), followed by ch-RCC (5%) [8]. Other types of RCC, such as collecting duct carcinoma or tubulocystic are very rare (1%). The remaining 4% of diagnosed cases belong to unclassified histopathological types of RCC [9,10]. The risk of RCC increases in the older population (especially between 60 and 70 years old) [11]. The other risk factors are smoking, hypertension, diabetes, obesity, reduction of glomerular filtration rate (GFR), and end-stage renal disease (ESRD) [12].

The most common RCC subtype is associated with a defect in the von Hippel-Lindau gene (*VHL*). Its product—the VHL protein, —promotes the proteolysis of the hypoxia-inducible factors 1a (HIF-1a) and 2a (HIF-2a). These proteins are the key regulators of the hypoxic response in cells. The loss of VHL function results in increased expression and stabilization of HIF, which is necessary for tumor growth in cancer cells [13]. It was shown that increased expression of HIF-1 is one of the strongest pro-angiogenic signals that promote tumor angiogenesis [14]. Additionally, this factor is responsible for the decreased expression of pro-apoptotic proteins (Bid and Bad) [15]. Furthermore, HIF-1 plays a role in the suppression of innate and adaptive immune responses against the tumor. It can induce the expression of immunosuppressive molecules such as immune check-point proteins, which are the therapeutic target in RCC [16]. Due to its functions, it seems that the inhibition of HIF may be beneficial in RCC. Indeed, FDA approved the use of belzutifan (a small molecule inhibitor of HIF-2α) in patients with RCC harboring the VHL mutation [17].

More than 50% of patients with RCC are asymptomatic. They are diagnosed incidentally during thoracoabdominal imaging ordered for other indications [18]. Due to the asymptomatic character of RCC, a lot of cases are detected too late. At the moment of diagnosis, 25% of patients have metastasis. Patients with 4th stage RCC have a 12% 5-year survival rate compared with 87% survival in patients with 1st stage RCC [19]. According to the stage of the disease, the recommended therapy should be applied. The first-line treatment in operatable RCC is surgical intervention. Partial nephrectomy is the method of choice in patients who have tumors measuring less than 7 cm. Radical nephrectomy is recommended for patients with bigger tumors. In cases of metastatic disease, due to chemoresistance, targeted molecular therapy should be administered. In clinical trials, anti-VEGF molecules (bevacizumab and sunitinib) were shown to be effective. Additionally, immune checkpoint inhibitors (e.g., pembrolizumab) were approved for patients with advanced RCC [20,21]. Because of the above-mentioned problems and poor outcomes for patients, new methods that ensure faster diagnosis or better disease control in patients with RCC are urgently needed. It seems that extracellular vesicles (EVs) can provide such an opportunity.

## 2. Extracellular Vesicles (EVs)

EVs are small lipid membrane vesicles, which are secreted by almost all kinds of cells into the extracellular matrix [22]. Based on their size and origin, we can distinguish between small EVs (sEVs; referred to as exosomes in the past) and large EVs (lEVs, referred to as microvesicles in the past). sEVs have an endosomal origin and a size less than 200 nm, while lEVs originate from the plasma membrane. Their size is larger than 200 nm (up to 1 µm) [23,24]. (Figure 1(1)).

EVs can be one of the most important ways of intracellular communication. They can affect cells at close or distant sites and modify the metabolism of targeted tissues [25]. Currently, we are constantly expanding our knowledge of EVs and their use in medicine, increasingly appreciating their predictive value. We observe their involvement in oncology and general medicine, e.g., cardiology and nephrology [26,27].

EVs can play a significant role in cancer development, growth, and metastasis formation [28,29]. The first information about potential chances to use EVs in new methods of oncological therapies was revealed in the 1990s. It was noted that B lymphocytes and dendritic cells secrete sEVs which are involved in immune response [30]. Exploiting electron microscopy, biochemistry, and functional assays, it was shown for the first time that antigen-presenting cells secrete sEVs that can stimulate T cell proliferation. This study using tumor-bearing mice reported that sEVs secreted by dendritic cells induced antitumoral immune responses in vivo [31]. The oncological aspect of lEVs has been less investigated than that of sEVs, but recently we can observe a growing amount of evidence that they are also involved in tumor development [32]. Moreover, cancer cells secrete more EVs than their nonmalignant counterparts [33]. Since EV release is a normal physiological process of all cells in development and homeostasis and is not oncogenic per se, the question has risen why especially cancer cells secrete such huge amounts of EVs and for what purpose. Hence, cancer researchers have joined the EV research field and contributed in a large part to its exponential growth. Thanks to their research we know today that both tumor-derived EVs as well as EVs derived from cancer-educated non-cancer cells, shape a tumor permissive environment and take part in many, if not all, steps of cancer progression. EVs induce the transformation of non-malignant cells within the tumor microenvironment (TME) and are responsible for the induction of changes in cells, which are named “hallmarks of cancer” [34,35,36]. Evidence suggests so far that the classic “hallmarks of cancer”, such as oncogenes and mutations, altered signaling, metabolic reprogramming, and microenvironmental status cause a shift towards EV release from cancer cells [37]. In this manuscript, we will focus on the role of EVs in the modulation of the immune system in patients with RCC.

## 3. Role of EVs in RCC Immune Escape

The role of EVs in the escape of RCC from tumor-specific immunity is under investigation. Recent studies suggest that tumor-secreted EVs can modulate the immune response by inhibition of T cells, NK response, and DC differentiation or by promoting myeloid-derived suppressor cells (MDSCs).

Yang et al. (2013) provided the first study about the potential role of RCC-derived EVs in the induction of immunosuppression. They found that EVs isolated from human kidney adenocarcinoma (ACHN) cells can inhibit T-cell proliferation by delivering Fas-ligand (FasL) on their surface (Figure 1(2)). It is a common mechanism of tumor-derived EVs in shaping anti-tumor immune competence [38]. It results in the induction of Fas receptor-mediated apoptosis which involves both extrinsic and intrinsic apoptotic pathways [39]. RCC-derived exosomes increase the expression of caspase-3, -8, and -9 and reduce the anti-apoptotic Bcl-2 in Jurkat T-cells. Moreover, they decrease the spontaneous release of cytokines by T-cells—interleukin-2 (IL-2), interferon-γ (IFN-γ), IL-6, and IL-10—which in effect can intensify the immunosuppression [40]. Next, Grange et al. (2015) demonstrated that EVs can block monocyte differentiation and maturation towards DCs in the RCC microenvironment and inhibit T-cell immune response (Figure 1(3)). These effects have been related to the expression of HLA-G on EVs released by tumor cells [41]. HLA-G is a non-classical MHC class I molecule that plays an important role in the modulation of the anticancer immune response [42]. Its expression is specifically up-regulated in primary RCC lesions obtained from patients. However, no correlation was observed between HLA-G expression and various clinicopathological parameters. [43]. HLA-G can affect the immune system in many ways and is a potential target for novel immunotherapies. Thus the role of EV-associated HLA-G may be not fully studied yet [44].

The potential effect of RCC-derived EVs on MDSCs has been demonstrated by Diao et al. (2015). In their study, exosomes isolated from Rencacell culture induced the suppressive activity of MDSCs by Hsp70-related p-Stat3 activation [45]. Stat3 is an important factor in immune escape, which signaling prevents differentiation of MDSCs into mature cell types and is involved in their suppressive activity and inhibition of the anti-tumor response (Figure 1(4)) [46]. The potential downstream mechanism of exosomal Hsp-70 immunosuppression is associated with increased expression of arginase-1 and iNOS (inducible NO synthase). The activity of these enzymes within MDSCs results in the inhibition of T-cells. Moreover, the Hsp70-induced expression of these proteins was reduced by a Stat3 inhibitor [45].

Next, it has been revealed that EVs can trigger dysfunction of NK cells in ccRCC patients. Xia et al. (2017) compared the NK cell function of ccRCC patients with benign tumor patients and healthy donors. Tumor-infiltrating NK cells of ccRCC patients presented an inhibitory phenotype and impaired function, which were associated with tumor progression. Exosomes from primary tumor cells of these patients were enriched in TGF-β1 (Transforming Growth Factor-β), which induces the TGF-β/SMAD pathway within NK cells. The SMAD (Suppressor of Mothers against Decapentaplegic) transcription factor regulates the proper activity of NK cells and its overexpression leads to their dysfunction (Figure 1(5)) [47,48]. Moreover, the authors showed that an exosome inhibitor can attenuate the inhibitory effect of ccRCC cells on NK cells, increasing their cytotoxicity.

EVs derived from RCC contain also other specific proteins, which immunosuppressive properties have been confirmed previously apart from EVs. For example, the expression of carbonic anhydrase 9 (CA9), CD70, and CD147 are upregulated during RCC development and progression. A recent study by Himbert et al. (2020) showed increased expression of CA9, CD70, and CD147 in cells and EVs compared to normal tissues and indicated these proteins as specific markers of EVs in ccRCC [49]. (Figure 1(2)).

CD70 is a unique ligand interacting with CD27 on the surface of T-cells, B-cells, and DCs that modulates their response. CD27 promotes T-cell activation and survival via NF-κB (nuclear factor kappa-light-chain-enhancer of activated B cells) and JNK (c-Jun N-terminal kinases) pathways. However, continuous stimulation caused by the strong expression of CD70 may disturb the proper regulation of immune response and induce an exhaustion effect of T-cells or even their apoptosis [50,51]. In normal conditions, expression of CD70 is restricted to antigen-stimulated immune cells and generally absent in non-lymphoid normal tissues [50,52]. In the case of cancer, we can observe high expression of CD70 in various malignancies, including both cc- and p-RCC, which is associated with its high abundance on both cell and tumor-derived EVs surface [49,53,54]. In ccRCC, it can be associated with dysregulation of the hypoxia-related pVHL/HIF pathway. Upregulation of the HIF level correlates with the increased expression of CD70 in these cells [51]. This effect is caused by the HIF-2α mediated inhibition of DNMT1 (DNA Methyltransferase 1), which increases the expression of CD70 [55]. Moreover, it was observed that hypoxic stress promoted EVs loading of the CD70 protein in lung cancer cells [56]. HIFs, hypoxia-inducible transcription factors, like HIF-1αand HIF-2α, are the main mediators of the tumor hypoxic response. Hypoxia is a pervasive feature of human tumors and is associated with poor prognosis and therapy resistance. It affects many aspects of tumor biology, however, in the context of this review, its key role in the generation of an immunosuppressive TME is the most interesting. Hypoxia-induced stabilization of HIF-1 has been shown to promote multiple immunosuppressive mechanisms within the TME. Examples are the promotion of Treg formation and recruitment, upregulation of PD-L1, CTLA-4, and LAG-3 to induce T-cell anergy and tolerance, recruitment of suppressive MDSCs and M2-like macrophages, the blocking of DC and NK-cell functions, suppression of T-cell activity through the accumulation of extracellular adenosine, etc., reviewed in [57].

CD70 mediates apoptosis of lymphocytes in RCC culture, which can be partially blocked with anti-CD27 and anti-CD70 antibodies [58]. Interestingly, strong CD70 expression in ccRCC and lymphocyte infiltration correlates significantly with worse patient survival compared to CD70-negative tumors without lymphocyte infiltration [51]. EVs derived from CD70-positive RCC may deliver CD70 and play an important role in the durable interaction of RCC and T-cells. (Figure 1(2)) It may be an important mechanism of RCC immune escape, increasing the malignancy of the tumor.

CA9 is an enzyme that catalyzes the hydration of carbon dioxide to carbonic acid, expelling protons to the extracellular matrix. Therefore, it is in charge of intracellular pH regulation of tumor cells (maintaining a neutral/alkaline intracellular pH) increasing at the same time extracellular acidity, which has distinct effects on the immune response and plays an important role in the suppression of anticancer immunity within the TME [59,60]. For example, the analysis of expression and clinicopathological data indicates a negative correlation of CA9 expression with CD8+ T cell infiltration and a positive correlation of these two parameters with the outcome in pancreatic cancer patients [61]. This effect may be caused by the CA9-mediated induction of the extracellular acidic microenvironment that inhibits cytolytic CD8+ T cells, as observed in vitro [61]. Increased CA-9 expression is also associated with FOXP3+ T reg abundance in the microenvironment of non-small-cell lung carcinomas and correlates with poorer survival of patients (*p* < 0.0001, HR 1.95, 95% CI: 1.3–2.7) [62]. Circulating lEVs carrying CA9 have been shown to be a potential factor in the diagnosis and prognosis of the ccRCC [63]. Therefore, the role of EVs-dependent delivery of CA9 in the regulation of RCC TME acidity and immune escape should be further studied (Figure 1(2)).

The metabolic interplay between tumor cells and tumor-infiltrating lymphocytes (TILs) can be also altered by CD147, which contributes to cancer immune escape by modulation of tumor glycolysis and the transport of monohydrates [64]. CD147, also known as extracellular matrix metalloproteinase inducer (EMMPRIN) or basigin (BSG), is a glycoprotein initially known as a regulator of matrix metalloproteinases. In addition to its well-described non-metabolic molecular mechanisms responsible for tumor progression, CD147 is recently recognized as a key factor in glucose metabolism reprogramming of tumors, resulting in a competition for nutrients between tumor cells and TILs in the TME [64]. CD147 expression by cancer cells causes inhibition of T-cell function also by exploiting the stimulatory ligand cyclophilin A (Figure 1(2)) [65]. In their meta-analysis, Li et al. (2017) observed a strong relationship between CD147 expression in RCC patients with higher TNM stage, histopathologic stage, lymph node metastasis, and worse 5-year survival compared with negative patients (HR = 1.61, 95%CI = 1.04–2.49) [66]. Moreover, they noticed no differences in the CD147 expression between clear cell RCC and other RCC types. Recently Chen et al. (2021) showed that CD147 expressed on TILs regulates antitumor CD8+ T-cell responses to facilitate the tumor-immune escape of mouse melanoma and lung cancer, and its deletion increases the abundance of TILs and cytotoxic function of CD8+ T cells [67]. In the same study, increased expression of CD147 in exhausted CD8+ TILs has been observed in the database of tumor biopsies [67]. Therefore, CD147+ EVs released by RCC cells into the TME may have a great impact on the activity of T-cells. Inhibition of EV-related delivery of CD147 across the TEM could therefore enhance the anti-tumor response.

Additionally, it was shown that EVs can play a role in immune escape by transferring PD-L1 (Programmed cell death protein ligand 1). It is a transmembrane protein responsible for the modulation of the immune response. By binding to its receptor PD-1 on the T-cell surface it can inhibit their cytotoxic activity and keep the immune response in check. In the case of cancer, tumoral PD-L1 expression is an important mechanism of immune escape. PD-L1-transfer via EVs derived from T-cells inhibits their function and plays a key role in evading the immune response. Poggio et al. (2019) showed that this effect is not restricted to the local tumor site but may also occur, due to EV-mediated transport, in the lymph nodes in a syngeneic mouse model of prostate cancer [68]. Wang et al. (2020) observed that an increased level of EVs with PD-L1 on their surface contributes to the suppression of T lymphocytes in patients with thyroid cancer. Moreover, patients with a higher level of PD-L1 EVs were characterized by a more advanced disease according to histologic specimens [69]. The PD-L1 level in the circulation turned out to be a marker of local and systemic immunosuppression and patient prognosis and response to treatment [38]. Immunotherapy targeting PD-1/PD-L1 has been applied in the clinic with the remaining issues of drug resistance, and EVs PD-L1 may play a role in mediating this effect. Therefore, considering immunotherapy to be a recommended treatment in RCC, the role of EVs-associated PD-L1 in immune escape and drug resistance in RCC may be crucial [70]. 

Besides proteins, EVs may improve the ability of tumors to avoid immunosurveillance by delivering also specific miRNA cargo. A miRNA is a short non-coding RNA of about 22 nucleotides that mediates gene silencing by guiding Argonaute proteins to target sites in the 3′untranslated region of target mRNAs [71]. RCC tissue samples show increased expression of miR-210 and 23a, which are also elevated in EVs isolated from the serum of RCC patients [72,73]. Their levels show a significant correlation with tumor staging and patients’ survival [72,73,74,75,76] Tumor-derived EVs can transfer them to neighboring cells, including immune cells [77]. Noman et al. (2012) showed that hypoxia-induced overexpression of miR-210 downregulates the expression of PTPN1, HOXA1, and TP53I11 genes, which decreased the susceptibility of non-small cell lung carcinoma cells to lysis by cytotoxic T cells in an in vitro study (Figure 1(6)) [78]. In another study in a mouse model, they observed enhanced MDSC-mediated T-cell suppression induced by miR-210 [79]. Hypoxia-induced miR-210 regulated MDSC function by selectively increasing arginase activity and NO production. Hypoxic conditions also increase the formation of tumor-derived lEVs by increasing RAB22A expression [80]. Moreover, hypoxic tumor-derived lEVs (referred to as MVs in the study) show a significantly increased expression of miR-210 and miR-23a [77]. These EVs can be taken up by NK cells, where miR-23a directly targets the expression of the CD107a—lysosomal-associated membrane protein. That results in decreasing their antitumor immune response (Figure 1(5)) [77]. MiR-23a is also a repressor of the transcription factor BLIMP-1, which promotes CD8+ T lymphocyte cytotoxicity and effector cell differentiation. Increased expression of miR-23a inside these lymphocytes alters their functions, increasing immunosuppression in the tumor environment (Figure 1(2)) [81]. Silencing of tumor-associated NK cells is also a property of miR-183, which promotes the proliferation and metastasis of renal cell carcinoma [82,83]. MiR-183 binds and represses DNAX activating protein 12 kDa (DAP12), an important signal adaptor required for proper NK cytotoxic activity (Figure 1(5)). The expression of miR-183 is increased in EVs released from human renal cancer stem cells and its level is upregulated in blood and tissue samples of RCC patients [82,84,85,86]. 

## 4. EVs as RCC Biomarker and Prognostic Factor

Late diagnosis is one of the main reasons for the high mortality rate among patients with RCC. Metastasis is often found at initial diagnosis or even after curative treatment [87]. For this reason, the need for sensitive and specific biomarkers is highlighted in most of the RCC studies. EVs may be a part of this revolution. EVs can be isolated from body fluids, so they have a significant potential to become important biomarkers and prognostic factors in medical oncology. However, the problem with proper isolation and characterization of EVs may be an obstacle in adopting this revolution to daily practice. It will need time and further intense investigations before EV-based biomarkers can be commonly used in medicine. A summary of transitional studies that investigated the content and the diagnostic/prognostic potential of EVs isolated from body fluids patients with RCC is provided in the table (Table 1). 

In some cases, the authors have revealed that EVs differ between patients and healthy controls [88]. Moreover, it has been shown that the content of RCC-derived EVs may be associated with the histologic subtype and can be a useful tool in early diagnosis and in the assessment of the cancer stage. Additionally, it was shown that EVs can be used as a prognostic factor and could predict the risk of having microvascular invasion revealed by pathological examination of surgically resected specimens [89]. In the following section, we have decided to focus on studies based on patients’ material, for their higher significance for the medical practice. Additionally, we presented the potential clinical applications of these findings for RCC immunology which can be developed based on the findings of basic studies performed not only on RCC but also on other cancer cell lines. The most relevant results of basic science studies performed on RCC cell lines, which paved the way for further patient studies, have been partially discussed in the previous section about the role of EVs in RCC immunology. Some results from basic studies have already been adopted for clinical practice. For example, Zieren et al. performed a multi-omic analysis on RCC and benign kidney cell lines. They identified 34 proteins as a candidate for pRCC biomarkers in EVs, and 20 proteins and 8 mRNA for ccRCC [90]. The results of this basic research pushed them to verification of their hypothesis in a human study. They showed that four tumor-specific mRNA (ALOX5, RBL2, VEGFA, TLK2) present in EVs (defined as present in tumor-derived-, urine- and plasma-EVs, but absent in tumor-adjacent tissue EVs) may be used as RCC biomarkers in the future (Table 1). The authors enrolled in their study only 11 patients with ccRCC before radical nephrectomy, due to the low number of patients with other histological types. They analyzed EVs isolated from urine, blood samples, tumor tissue, and tumor-adjacent tissue (obtained from pathological specimens) [91]. Interestingly, none of these four mRNA molecules have shown a significant difference in levels in their basic research on kidney benign and cancer cell lines. However, the reason for this discrepancy may be the different sources (cell media in a basic study or urine in a clinical study) from which EVs were isolated. Only one mRNA (STAT1) was detected in plasma EVs, so the authors decided to exclude plasma from the analysis [91]. The potential impact of the proteins coded by these mRNAs on the immune system is provided in the table (Table 1). 

As it was mentioned before, EVs can be useful not only for diagnosis but also for the staging of RCC. Horie et al. observed in cell culture experiments an increased activity of exosomal γ-glutamyl transferase (GGT) in the cell culture supernatant from neoplastic cells in comparison to the supernatant of normal ones. Additionally, they observed higher activity of the exosomal fraction of this enzyme in patients with advanced tumor stages (III and IV vs. I and II, *p* = 0.037). They also noted that this activity is positively correlated with microvascular invasion of RCC. Interestingly, these results were not observed in serum GGT (non-exosomal fraction) [89]. More examples of the usage of EVs as a diagnostic and prognostic factor are presented in the table (Table 1). The table also summarizes the potential impact of the given EV content on the immune system and its role in tumor RCC development (Table 1 and Figure 2).

## 5. Future Perspective

RCC is a chemotherapy and radiotherapy-resistant tumor. Immunotherapy and target treatment (e.g., tyrosine kinase inhibitors) are the basis of available treatment protocols. However, some patients do not respond to the applied therapy, so new therapeutic options or methods which can increase the vulnerability of the tumor to used treatments are needed. The role of EVs in oncological therapy is still not well investigated. However, recently an increasing number of researchers have been conducting studies investigating the hypothesis that EVs may be crucial in tumor homeostasis.

It was observed that sEVs can be a mediator of drug resistance in renal cell cancer. Qu et al. showed that the long non-coding RNA lnc-ARSR can compete with endogenous miR-34 and miR-449, and therefore increases the expression of their target genes (*AXL* and *c-MET*). These genes can be essential in sunitinib resistance development. Indeed, it was confirmed that an increased level of lnc-ARSR was negatively correlated with sunitinib response. Additionally, the authors have proven that the sEVs are the main way of intracellular transfer of lnc-ARSR between resistant and sensitive cells. Moreover, it was shown that the inhibition of sEVs formation or the packing of lnc-ARSR into them can be a method to restore sunitinib response [92]. Also, another research group observed that the inhibition of EVs biogenesis (by a selective inhibitor or ketoconazole) can potentiate the efficacy of sunitinib [93]. Another resistance mechanism mediated by EVs was revealed by He et al. [94]. They showed that EVs can carry miR-31-5p, which downregulates the MutL homolog 1 (MLH1) and therefore promotes sorafenib resistance. The resistant cells release EVs enriched in this miRNA [94]. The use of antagonists against lnc-ARSR or miR-31-5p may result in higher sensitiveness for these drugs used in target therapy in patients and can result in better outcomes in patients. However, this hypothesis should be further evaluated in clinical trials. Interestingly, sunitinib can modulate the anti-tumor immune response. It was shown that sunitinib therapy significantly increases T cells activity defined as enhanced proliferation after exposure to phytohemagglutinin and tetanus toxin. Additionally, a lower number of T regulatory cells was detected in patients’ blood after this therapy [95]. 

The EVs may be also used in anti-tumor therapy. Zhang et al. showed that EVs derived from genetically modified RCC cells may increase the activation of T cytotoxic lymphocytes and the production of IFN-gamma [96]. The modification consisted of the addition of a plasmid into the RCC cells (cell line RC-2) for the enrichment of the EV membrane in IL-12 (sEVs/IL-12). The analysis revealed that the amount of released IFN-γ was the highest (174.3 ± 14.6 pg/mL, *p* < 0.05) when T lymphocytes were stimulated by sEVs/IL-12 in comparison to PBS and non-modified EVs. Additionally, the usage of sEVs/IL-12 in research trials resulted in the highest killing rate and antigen-specific cytotoxic effect against cancer cells [96]. The potential of sEVs to stimulate T lymphocytes was also confirmed by Xu et al. [97], however, only in cell culture and further investigation is needed to check the safety and efficacy of this procedure in vivo. 

Another research group noted that tumor-derived EVs effectively stimulate dendritic cell (DC) activation and upregulate CD11c, MHCII, and IL12 expression in DC. The use of DC loaded with tumor-derived (from RENCA cell line) EVs in a mice model of RCC significantly prolonged the survival time of the animals. Additionally, these EVs strongly promoted the recruitment of CD4^+^ and CD8^+^ T lymphocytes. The authors also conducted an experiment showing that intravenous injection of these EVs does not induce tumor immune tolerance in animals [129]. Recently, Greenberg et al. published an article that shows that tipifarnib (farnesyltransferase inhibitor) can inhibit EV biogenesis. Additionally, the authors proved in cell culture that the expression of PD-1L on the EV surface, which is responsible for the cytotoxic effect of these EVs on lymphocytes, is lower after treatment with tipifarnib and sunitinib [130]. Furthermore, the EVs can also be a part of an oncology treatment that is directly and exclusively associated with the stimulation of the immune system. In one study, authors observed in an RCC (A498) cell line that the injection of garlic (*Allium sativum*)-derived sEVs can inhibit tumor cell proliferation and induce their apoptosis [131]. Additionally, it was shown that circular RNA_400068 transferred by sEVs can play a role in RCC pathogenesis by decreasing the level of miR-210-5p. This miR downregulates the expression of the suppressor of cytokine signaling 1 (SocS1). Additionally, as it was mentioned above, mir-210 may regulate MDSC suppressor function by increasing arginase activity and NO production. In consequence, treatment of healthy kidney cells with exosomal circRNA_400068 increased the proliferation and decreased the apoptosis rate of these cells, representing therefore a new oncogenic factor [132]. Another research group noted that a higher level of miR-15a, which downregulates the expression of the B-cell translocation gene 2 (BTG2), promoted the proliferation, migration, and invasion of ccRCC cells. They postulated that inhibition of this miR can be used as a part of therapy in the future [133].

Furthermore, it was noted that EVs can take part in the progression of metastatic disease. It seems that they can be a target in the treatment of advanced RCC. Jin et al. observed that EVs derived from RCC cells (cell line 786-O) can promote lung metastasis formation and tumor growth. They postulate that these effects are induced by MALAT-1. Potentially, the suppression of this transcript factor can reduce cells’ malignant behavior [134]. All this evidence confirms that EVs should be considered as part of immunotherapy in patients with RCC, after confirmation of their safety and efficacy in clinical trials.

## 6. Conclusions

The results of both experimental and clinical studies indicate that EVs are one of the key players in the immune response against RCC. The presence and content of EVs can induce immunosuppression that can be crucial in cancer development. Additionally, the molecular cargo of EVs may have a negative impact on intercellular signaling in the surrounding tissue and may reshape the TME towards facilitating tumor growth. 

Based on this information, it can be concluded that EVs are involved in resistance to treatment of RCC. They can decrease the sensitiveness for immune-check-point inhibitors in patients and block the stimulation of lymphocytes. Moreover, EVs may be a new therapeutic option in the future, due to positive results of in-vitro and in-vivo preclinical studies. Perhaps the detection of circulating EVs for the prediction and prognosis of oncological patient outcomes could have greater clinical implications in the future. However, randomized controlled trials are needed to verify these hypotheses.

## Figures and Tables

**Figure 1 jpm-12-00772-f001:**
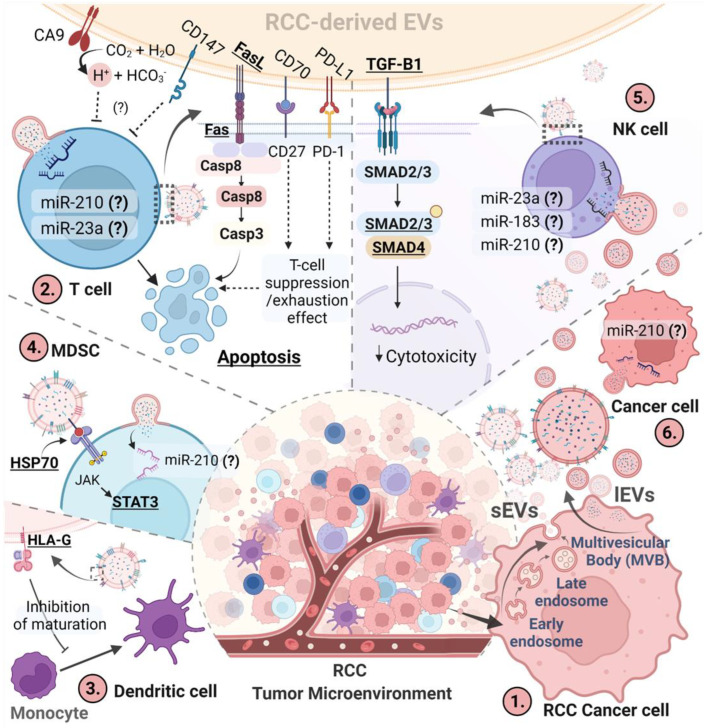
The role of EVs in immune escape of RCC; (?)—potential immunosuppressive mechanism not confirmed in the context of RCC-derived EVs but described in other cancer types. RCC cancer cells secrete small EVs (sEV) through the endocytic pathway or large EVs (lEVs) by membrane blebbing (**1**.). EVs inhibit T-cell proliferation and induce apoptosis by delivering FasL, cause T-cell exhaustion and apoptosis through CD70, and inhibit T-cell function by creating an acidic microenvironment through the enzyme CA9 and through the delivery of CD147 or PD-L1. EV-associated miR-23a represses through the transcription factor BLIMP-1, CD8+ T lymphocyte cytotoxicity, and effector cell differentiation (**2**.). EVs block monocyte differentiation and maturation by delivery of HLA-G. (**3**.). EVs induce the suppressive activity of MDSCs by Hsp70-related p-Stat3 activation EV-delivered miR-210 up- regulates MDSC suppressor function by increasing arginase activity and NO production (**4**.). EVs cause dysfunction of NK cells by delivery of TGF-B1, which induces the TGF-β/SMAD pathway. EV-associated miR-23a suppresses NK function by blocking the expression of CD107a and EV-delivered miR-183 binds and represses DAP12, an important signal adaptor required for proper NK cytotoxic activity (**5**.). Hypoxia induces overexpression of miR-210 in RCC cancer cells, which downregulates PTPN1, HOXA1, and TP53I11 genes and decreases susceptibility to cytotoxic T-cell lysis (**6**.). Created with BioRender. Abbreviations: BLIMP-1-B lymphocyte-induced maturation protein-1; CA-9—Cancer antigen 9; Casp—caspase; CD—Cluster of differentiation; CO_2_—carbon dioxide; EVs—extracellular vesicles; FAS—Fas Cell Surface Death Receptor; FasL—Fas Cell Surface Death Receptor Ligand; H+—hydrogen ion; HCO_3_—bicarbonate anion; HLA—human leukocyte antigen; H_2_O—water; HSP—heat shock protein; JAK—Janus-activated kinases; lEVs—large extracellular vesicles; MDSC—Myeloid-derived suppressor cell; miR—microRNA; MVB—Multivesicular body; NK—natural killers; PD-1—Programmed cell death protein 1; PD-L1—Programmed cell death protein ligand 1; RCC—renal cell carcinoma; sEVs—small extracellular vesicles; SMAD—Suppressor of Mothers against Decapentaplegic; STAT—Signal transducers and activators of transcription; TGF-β—Transforming Growth Factor- β.

**Figure 2 jpm-12-00772-f002:**
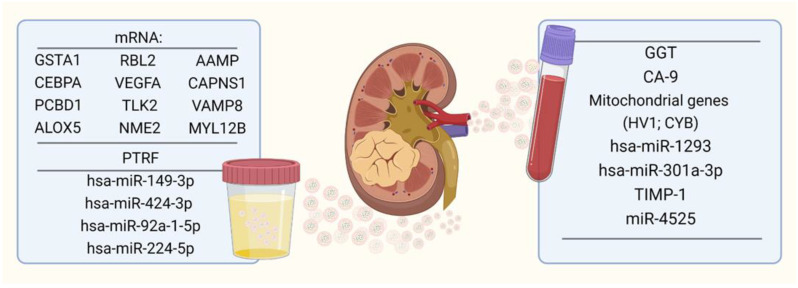
EV-associated biomarkers and prognostic factors with potential impact on the immune system of RCC patients, found in urine and blood. Created with BioRender. Abbreviations: AAMP—Angio-associated migratory cell protein; ALOX5—Arachidonate 5-lipoxygenase; CA-9—Cancer antigen 9; CAPNS1—Calpain small subunit 1; CEBPA—CCAAT/Enhancer binding protein alpha; CYB—Cytochrome B; EV—extracellular vesicle; GGT—Gamma-glutamyl transferase; GSTA1—Glutathione S-transferase A1; hsa-miR—human microRNA; HV1—Hirudin variant 1; miR—microRNA; MYL12B—Myosin light chain 12B; NME2—Nucleoside diphosphate kinase B; PCBD1—Pterin-4-alpha-carbinolamine dehydratase 1; PTRF—Polymerase and transcript release factor; RCC—renal cell carcinoma; RBL2—Retinoblastoma-like protein 2; TIMP-1—Tissue inhibitor of metalloproteinase 1; TLK2—Tousled-like kinase 2; VAMP8—Vesicle associated membrane protein 8; VEGF—Vascular endothelial growth factor-A.

**Table 1 jpm-12-00772-t001:** List of selected studies focused on EVs as biomarkers or prognostic factors with a description of their potential impact on the immune system.

Number of Patients	Investigated Molecule in EVs	Observed Correlation	Potential Role in Cancer Immunology	Sample	Reference
28 (20 with cc-RCC, 2 with p-RCC, 4 with ch-RCC, 2 with unknown type)	GGT	Higher level and activity of GGT in EVs in patients with advanced disease and microvascular invasion	May take part in T lymphocytes activation [98]	Serum	Horie et al. [89]
70 with cc-RCC	miR-30c-5p	Biomarker of cc-RCC (downregulated in comparison to 30 HI)—sensitivity and specificity were 68.57% and 100%, respectively	miR-30c-5p directly inhibits *HSPA5*, which can be negatively associated with ferroptosis and autophagy activation [99,100]. *HSPA5* is involved in UPR, which is important during tumor antigen presentation [101]	Urine	Song et al. [88]
4 with cc-RCC	PTRF	Increased expression and secretion of PTRF in comparison to HI	PTRF can be associated with higher expression of PD-L1 [102]	Urine	Zhao et al. [103]
29 with cc-RCC	Proteomic analysis	72/333 proteins were present only in cc-RCC patients (not in 23 HI)	Complement components and antibodies chains present in EVs may be evidence of immune response against tumor	Urine	Raimondo et al. [104]
22 (18 with cc-RCC, and 4 with p-RCC)	mi-RNA profile	hsa-miR-149-3p and hsa-miR-424-3p were upregulated; hsa-miR-92a-1-5p were down-regulated	miR-149-3p can promote EMT [105], miR-424 can decrease PD-L1 expression [106], miR-92a may be involved in IL-6 secretion [107]	Urine	Xiao et al. [108]
77 with cc-RCC	CA-9	Biomarker and prognostic factor of survival and recurrence	Expression of CA-9 can be associated with higher expression of PD-L1 [62]	Plasma	Vergori et al. [63]
6 with cc-RCC	miRNA profile	Increased level of miR-224-5p	miR-224-5p inhibits the expression of CCND1, which increases PD-L1 protein abundance [109]	Urine	Qin et al. [109]
13 (12 with cc-RCC, 1 with p-RCC)	Mitochondrial genes *HV1; CYB*	Detection of metastasis and aggressiveness	They are a marker of increased ROS production which can modulate T lymphocytes survival [110]	Plasma	Arance et al. [111]
32 with cc-RCC	hsa-miR-301a-3p; hsa-miR-1293	Decrease of hsa-miR-1293 and increase of hsa-miR-301a-3p were the biomarker of metastatic disease	miR-1293 regulates the expression of proteins involved in DNA repair processes [112], miR-301a is involved in T-lymphocytes accumulation and IFN-gamma production [113]	Plasma	Dias et al. [114]
32 with localized cc-RCC and 23 with metastatic cc-RCC	TIMP-1	Biomarker of tumor size and presence of metastasis	TIMP-1 can trigger NET formation [115], induces sensitivity to FAS-related apoptosis of cancer cells [116]	Plasma	Dias et al. [117]
33 with cc-RCC	mRNA	Decreased levels of *GSTA1*, *CEBPA,* and *PCBD1* mRNA	CEBPA can be involved in tumor-induced immunosuppression (by regulating the function of MDSCs) [118], GSTA1 is involved in ROS production, which can promote tumorigenesis and regulate T lymphocytes function [119,120], PCBD1– ND	Urine	De Palma et al. [121]
6 with cc-RCC	mRNA	Decreased levels of *NME2*, *AAMP*, *CAPNS1*, *VAMP8*, and *MYL12B* mRNA	Nme2 can stimulate Tc lymphocytes [122], Aamp can polarize M population into M1 [123], Myl12b may be the ligand for CD69 (suppressor of anti-tumor immune response) [124], Vamp8, Capns1 ND	Urine	Marek-Bukowiec et al. [125]
8 with cc-RCC	miRNA	Elevated level of miR-4525	ND	Serum	Muramatsu-Maekawa et al. [126]
9 with cc-RCC	mRNA	Presence of four types of mRNA (ALOX5, RBL2, VEGFA, TLK2) was specific for cc-RCC patients	ALOX-5 expressed by macrophages can be a precursor of pro-tumorigenic metabolites [127], RBL2, TLK2 regulate the cell division process [91], VEGFA pathway can stimulate the proliferation of tumor-induced T-regulatory lymphocytes [128]	Urine	Kuczler et al. [91]

AAMP—Angio-associated migratory cell protein; ALOX5—Arachidonate 5-lipoxygenase; CA-9—Cancer antigen 9; CAPNS1—Calpain small subunit 1; cc-RCC—clear cell renal cell carcinoma; CCND1—Cyclin D1; CD—Cluster of differentiation; CEBPA—CCAAT/Enhancer binding protein alpha; ch-RCC—chromophobe renal cell carcinoma; CYB—Cytochrome B; DNA—Deoxyribonucleic acid; EMT—Epithelial-mesenchymal transition; EVs—Extracellular vesicles; FAS—Fas Cell Surface Death Receptor; GGT—Gamma-glutamyl transferase; GSTA1—Glutathione S-transferase A1; HI—Healthy individuals; hsa-miR—human microRNA; HSPA5—Heat shock protein A5; HV1—Hirudin variant 1; IFN—interferon; IL—interleukin; M—macrophages; M1—macrophage type 1; MDSCs—Myeloid derived suppressor cells; miR—microRNA; MYL12B—Myosin light chain 12B; NET—Neutrophil extracellular traps; NME2—Nucleoside diphosphate kinase B; p-RCC—papillary renal cell carcinoma; PCBD1—Pterin-4-alpha-carbinolamine dehydratase 1; PD-L1—Programmed death-ligand 1; PTRF—Polymerase and transcript release factor; RBL2—Retinoblastoma-like protein 2; ROS—Reactive oxygen species; TIMP-1—Tissue inhibitor of metalloproteinase 1; TLK2-Tousled-like kinase 2; UPR—Unfolded protein response; VAMP8—Vesicle associated membrane protein 8; VEGFA—Vascular Endothelial Growth Factor A.

## Data Availability

Not applicable.

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
