# Peer review of "Extracellular Vesicles—A New Potential Player in the Immunology of Renal Cell Carcinoma"

_jpm, 2022, doi:10.3390/jpm12050772_

Round 1

Reviewer 1 Report

-some sentences are very big, try breaking down into smaller ones.

-Insert figures/ data from some of the cited work.

-In table 1, what are the references provided in column- potential role in cancer immunology?

Author Response

We thank the reviewer for carefully reading our manuscript and for the comments and suggestions to improve the overall quality of the work. Comments are answered point-by-point:

-some sentences are very big, try breaking down into smaller ones.

We have done so wherever it was appropriate (changes marked in the revised manuscript in red).

-Insert figures/ data from some of the cited work.

Where we found it appropriate, we discussed the cited work in more detail (changes in red), however we decided against adding new figures, since we did not wanted to overload the already long manuscript and we think that the existing figures are complex enough.

-In table 1, what are the references provided in column- potential role in cancer immunology?

We have inserted the appropriate references.

Reviewer 2 Report

The manuscript is well organized and highlights the relevance of EVs in RCCs, as well as their role in anti-cancer treatment.

The manuscript is written in correct English, is complete and clear and does not require minor revisions. The references are also recent and there is no need for integration.

There are no other comments to add.

Best regards

Author Response

We thank the reviewer the positive comments and appreciate his recognition of our work.

Reviewer 3 Report

Extracellular vesicles are increasingly being recognized as important players in cancer and this review summarizes the current field with a special view on renal cancer and the immunology thereof.

While the review is very interesting and contains a lot of information there should be some more work done to give the text more structure, clarify paragraphs to make it easier for the reader to put things in context.

Specific comments are provided below.

  • Some sections read like a list of facts but not really a coherent story (e.g. CA9 section line 136-149). The authors should try put each component they discuss in context with their main biological function.
  • Briefly explain what the functions of CD70, CA9, CD147, PD-L1….. are at the beginning of each section. Similar to how it is done for CA9 but with more detail.
  • Given the quite prominent role for HIF in the regulation of many of the components associated with modulation of immune response in the microenvironment through EV’s the authors should introduce HIF in more detail.
  • Introducing the cause for release of EV’s from cancer cells in the intro would be very interesting.
  • What is the (?) in figure legend 1?
  • Line 22 – “try” doesn’t seem to do your review justice
  • Line 32/33 what changed in the kidney cancer treatment that caused the decrease in mortality?
  • Line 36ff: RCC types don’t add up to 100%, what are the remaining 5%?
  • Standard of care of RCC should be mentioned in the introduction section.
  • Line 69: 1990s
  • Line 117: referring to CD70 as particle is a bit odd
  • Line 125: There are several reports suggesting that CD70 is a HIF target gene. Mention here to aid the mechanistic understanding (see also comment above).
  • Line 137: tumor microenvironment refers to the extracellular milieu around the tumor. What do the authors mean by “intracellular pH of the microenvironment”? Does CA9 influence extracellular or intracellular acidification?
  • The figure is too packed. The authors should consider separating it into smaller sub panels.

Author Response

We thank the reviewer for carefully reading our manuscript and for the comments and suggestions to improve the overall quality of the work. Comments are answered point-by-point (changes marked in the revised manuscript in red):

Some sections read like a list of facts but not really a coherent story (e.g. CA9 section line 136-149). The authors should try put each component they discuss in context with their main biological function.

Briefly explain what the functions of CD70, CA9, CD147, PD-L1….. are at the beginning of each section. Similar to how it is done for CA9 but with more detail.

We have explained the biological functions, especially in the context of immunosuppression, in more detail for every mentioned molecule at the beginning of each section: FasL (lines 120-126), HLA-G (lines 127-132),, CD70 (lines 161-168), CA9 (lines 193-200), CD147 (lines 211-216), PD-L1 (lines 231-238).

Given the quite prominent role for HIF in the regulation of many of the components associated with modulation of immune response in the microenvironment through EV’s the authors should introduce HIF in more detail.

We have done so in lines 51-59 and 175-185.

Introducing the cause for release of EV’s from cancer cells in the intro would be very interesting.

We have mentioned this topic in the paragraph lines 64-72

What is the (?) in figure legend 1?

We have explained it more clear in the figure legend

Line 22 – “try” doesn’t seem to do your review justice

We have changed it.

Line 32/33 what changed in the kidney cancer treatment that caused the decrease in mortality?

We have added an explanation in lines 33-36

Line 36ff: RCC types don’t add up to 100%, what are the remaining 5%?

We have added an explanation in lines 41-43

Standard of care of RCC should be mentioned in the introduction section.

We mentioned it in lines 64-72.

Line 69: 1990s   We have corrected it.

Line 117: referring to CD70 as particle is a bit odd   We have corrected it.

Line 125: There are several reports suggesting that CD70 is a HIF target gene. Mention here to aid the mechanistic understanding (see also comment above).

We have mentioned it in lines 171-175.

Line 137: tumor microenvironment refers to the extracellular milieu around the tumor. What do the authors mean by “intracellular pH of the microenvironment”? Does CA9 influence extracellular or intracellular acidification?

We have explained it more clearly in lines 194-197.

The figure is too packed. The authors should consider separating it into smaller sub panels.

We separated the initial figure into 2 separate figures 1 and 2.